# Early Nutritional Intervention to Promote Healthy Eating Habits in Pediatric Oncology: A Feasibility Study

**DOI:** 10.3390/nu14051024

**Published:** 2022-02-28

**Authors:** Véronique Bélanger, Josianne Delorme, Mélanie Napartuk, Isabelle Bouchard, Caroline Meloche, Daniel Curnier, Serge Sultan, Caroline Laverdière, Daniel Sinnett, Valérie Marcil

**Affiliations:** 1Department of Nutrition, Université de Montréal, Montreal, QC H3T 1A8, Canada; v.belanger.7@gmail.com (V.B.); josianne.delorme@umontreal.ca (J.D.); melanie.napartuk@umontreal.ca (M.N.); 2Research Centr, CHU Sainte-Justine, Montreal, QC H3T 1C5, Canada; isabelle.bouchard.hsj@ssss.gouv.qc.ca (I.B.); cmeloche.hsj@gmail.com (C.M.); daniel.curnier@umontreal.ca (D.C.); serge.sultan@umontreal.ca (S.S.); daniel.sinnett@umontreal.ca (D.S.); 3School of Kinesiology and Physical Activity Sciences, Université de Montréal, Montreal, QC H3T 1J4, Canada; 4Department of Psychology, Université de Montréal, Montreal, QC H2V 2S9, Canada; 5Division of Hematology-Oncology, CHU Sainte-Justine, Montreal, QC H3T 1C5, Canada; caroline.laverdiere@umontreal.ca; 6Department of Pediatrics, Université de Montréal, Montreal, QC H3T 1C5, Canada

**Keywords:** childhood cancer, feasibility study, nutritional intervention, acute treatment phase

## Abstract

This study aims to describe the feasibility of a nutritional intervention that promotes healthy eating habits early after cancer pediatric diagnosis in patients and their parents. Participants were recruited 4 to 12 weeks after cancer diagnosis as part of the VIE study. The one-year nutritional intervention included an initial evaluation and 6 follow-up visits every 2 months with a registered dietician. The feasibility assessment included rates of retention, participation, attendance, completion of study measures, and participants’ engagement. A preliminary evaluation of the intervention’s impact on the participants’ dietary intakes was conducted. A total of 62 participants were included in the study (51.6% male, mean age = 8.5 years, mean time since diagnosis = 13.2 weeks). The retention and attendance rates were 72.6% and 71.3%, respectively. Attendance to follow-up visits declined over time, from 83.9% to 48.9%. A majority of participants had high participation (50.8%) and high engagement (56.4%). Measures of body-mass-index or weight-for-length ratio and dietary 24-h recalls were the procedures with the highest completion rates. Participants with refractory disease or relapse were less likely to complete the intervention. Post-intervention, participants (*n* = 21) had a lower sodium intake compared to the initial evaluation. These results suggest that a nutritional intervention that involves patients and parents early after a pediatric cancer diagnosis is feasible.

## 1. Introduction

In Canada, cancer remains the leading cause of death from disease in children and adolescents despite an almost 85% survival after 5 years [1,2]. Yet, childhood cancer survivors (CCS) are at a high risk of suffering from chronic health problems even as young adults [3,4], including cardiometabolic (CM) complications [5,6,7]. It is therefore recommended that CM parameters be closely monitored in children and adolescents who have been treated for cancer. Thus, both the American Heart Association (AHA) and the Children’s Oncology Group recommend the screening of blood pressure, blood lipids, glucose metabolism parameters, and body weight at the end of cancer treatment and to monitor their evolution yearly [8,9]. Moreover, like the general population, better healthy eating habits in CCS have been associated with better body fat parameters and CM health [10,11,12]. Therapeutic interventions for CCS presenting CM complications such as high body mass index (BMI), high blood pressure, and high LDL-C include the promotion of healthy lifestyle habits [8]. However, adopting healthy lifestyle habits as an adult can sometimes be challenging, and implanting them at an early age is undeniably a good strategy. 

Traditionally, nutritional and dietary intervention during cancer treatment in children and adolescents has been aimed at preventing or improving malnutrition [13,14]. The presence of malnutrition at diagnosis or during treatments has been associated with a lower tolerance to therapy, increased risk of infections, and worse outcomes [15,16,17]. The causes leading to poor nutritional status during pediatric cancer are multifactorial. Not only is cancer itself characterized as a state of metabolic stress that can lead to undernutrition, but treatments cause various pernicious digestive side effects such as dysgeusia, mucositis, nausea, vomiting, and diarrhea [13,14,18,19]. Dyslipidemia [20,21], glucose intolerance [22], high blood pressure [23], and an increase in BMI [18] are also acute side effects of cancer treatments. However, they have been poorly addressed by nutritional intervention during pediatric cancer therapy. A high BMI and obesity at diagnosis is another form of malnutrition associated with an increased risk of complications (i.e., wound infection and arterial thrombosis) [24] and, in acute lymphoblastic leukemia (ALL), of disease relapse [25]. 

Although obesity at diagnosis and increased BMI during treatment have been associated with CM complications in CCS [26,27], most of the nutritional interventions during pediatric cancer treatment have not considered the long-term impact of therapy. However, it has been proposed that patients should be educated about the importance of healthy eating and regular physical activity during cancer treatment [28,29,30,31]. The few lifestyle interventions for preventing long-term CM complications were developed solely for childhood ALL and implemented at the remission or maintenance phases [29,30,32,33]. Implementing such interventions early after cancer diagnosis brings challenges, given the significant medical, emotional, and financial difficulties encountered. Feasibility studies provide an opportunity to preliminarily assess the capacity to implement an intervention and its impact [34]. By documenting recruitment capacity, assessing intervention acceptability, and collecting study measures, these studies provide a flexible methodology that can be adapted during its course [34,35,36]. 

We propose that a nutritional intervention that begins shortly after diagnosis and involves patients and their parents is feasible. Here, we aim to describe the feasibility of a nutritional intervention promoting healthy eating habits early after cancer diagnosis and to investigate the relationships between feasibility data and participants’ characteristics (socio-demographic, cancer-related, and dietary intakes at the initial visit). Our secondary objective is to preliminary assess the impact of the intervention on participants’ dietary intakes.

## 2. Materials and Methods

### 2.1. Study Framework and Ethics

This work is part of the VIE (Valorization, Implication, Education) multidisciplinary study that consists of nutritional, psychological, and physical activity interventions at CHU Sainte-Justine (CHUSJ) in Montreal, QC, Canada. The VIE program aims at supporting and educating children diagnosed with cancer and their parents about healthy lifestyle habits to prevent short- and long-term health complications. The design of the physical activity and psychosocial interventions have been described in detail elsewhere [37,38]. The study was approved by the Ethics Review Committee of CHU Sainte-Justine (#2017-1413) and was in accordance with the Declaration of Helsinki.

### 2.2. Participants

Participants were recruited from February 2018 to December 2019. Inclusion criteria were: (1) being 21 years old or younger at diagnosis; (2) being treated with chemotherapy or radiotherapy; and (3) being able to give informed consent (by parents or legal guardians). The agreement of the attending oncologist was required for the participation of eligible patients. Patients were excluded if they were not being treated with chemotherapy and/or radiotherapy. Eligible participants were enrolled between the fourth and twelfth week after a cancer diagnosis. The physician had the right to withdraw participants from the study if their health condition no longer allowed them to participate.

### 2.3. Nutritional Intervention

The nutritional intervention aimed at promoting healthy eating behaviors to ensure normal growth and development of the child, weight maintenance during and after treatment, and prevention of long-term health complications. Participants were followed for at least 1 year from the initial nutritional evaluation. Follow-up visits were planned every 2 months, for a total of 6 during the first year. Two registered dieticians (RD) delivered the intervention to their assigned patients. At each visit, they provided individualized nutritional counseling addressing the side effects of cancer treatments, and encouraging changes in eating behaviors, confidence, self-esteem, assertiveness, and self-acceptance. The proposed approach favored repeated exposure to healthy food without pressure to eat and ensuring an eating routine [39,40,41]. Dietary restrictions and food classifications that induce dichotomous thinking of “good” and “bad” foods were proscribed as these methods have proven to be harmful and counterproductive in children [39,42]. Dietary intake and anthropometric profile were also monitored at each visit, and the RD encouraged the participant and/or parents to set specific dietary goals based on their needs.

The initial nutritional evaluation aimed at establishing the participant’s nutritional history and assessing their dietary needs. For this purpose, information on treatment side effects, weight history, and eating habits was gathered. Dietary intakes were collected with a 24-h recall (24H-R) and a 3-day food record that was handed out to be filled at home. Weight, height, waist circumference (WC), mid-upper arm circumference (MUAC), triceps skinfold thickness (TSFT), and subscapular skinfold thickness (SSFT) were measured. A blood test was planned to assess their nutritional status and cardiometabolic health (glucose, insulin, total cholesterol, high density lipoprotein (HDL)-cholesterol, low density lipoprotein (LDL)-cholesterol, triglycerides, glycated hemoglobin, C-reactive protein, and vitamin D).

At follow-up visits, the RD assessed the participant’s goal achievement and the need to adjust the nutritional objectives. Dietary intakes were collected with a 24H-R and a 3-day food record to be completed at home. Anthropometric measurements were repeated (weight, height, WC, MUAC, TSFT, and SSFT). After one year of nutritional intervention, the need for additional follow-up was discussed with the participant and/or parents and, if necessary, the frequency of appointments was set to their preference. Originally, meetings were planned to be delivered in person. However, between 15 March and 14 September 2020, meetings were conducted by phone due to the COVID-19 pandemic.

### 2.4. Participants’ Socio-Demographic and Cancer-Related Characteristics

Socio-demographic and cancer-related characteristics at recruitment were extracted from medical charts and case report forms. Data collected included medical information (diagnosis and diagnosis date), socio-demographic data of participants (age at diagnosis and age at recruitment) and families (parents’ marital status {married/common-law partners or separated/divorced/widower}, parental education level {less than high school, high school, college, or university} and approximative familial gross revenue {<$29,999, $30,000–$69,999, $70,000–$109,999, $110,000–$150,000, or >$150,000}). During the nutritional intervention, cancer evolution was documented via medical charts or communication with the healthcare providers. After one year of nutritional intervention, the patient’s cancer evolution was categorized as responder or non responder (patient with refractory disease or relapse).

### 2.5. Assessment of Nutritional Status and Vitamin D

Nutritional status at the initial evaluation was determined with BMI calculated as weight (kg)/height (m)^2^ (for patients aged from 2 to 19 years) or with a weight-for-length (W/L) ratio (for infants ≤2 years old) [43]. BMI-for-age and z-score of W/L ratio were obtained using the Microsoft^®^ Office Excel^®^ tool developed by the British Columbia Children Hospital and the Canadian Pediatric Endocrine Group based on the 2014 version of Growth Charts for Canada [44]. Determination of nutritional status was adapted to match Canada’s growth indicators statement [43]. Malnourished patients were determined by having a z-score <−2.0 of W/L ratio (≤ 2 years of age) and of BMI-for-age (>2 to 19 years old). For infants (0 to 2 years old) and children (2 to 5 years old), normal weight (≥−2.0 to ≤3.0 z-score) and obesity (>3.0 z-score) were determined using W/L ratio and BMI-for-age, respectively. In participants at least 5 years old, z-scores of BMI-for-age ≥−2.0 to ≤2.0 were considered normal weight while >2.0 were described with obesity [43,45,46]. 25-hydroxyvitamin D [25(OH)D] was measured in serum using liquid chromatography-tandem mass spectrometry. Vitamin D status was assessed based on the 2016 consensus report of experts that defines sufficiency: >50 nmol/L; insufficiency: 30–50 nmol/L; and deficiency: <30 nmol/L [47].

### 2.6. Evaluation of Study Feasibility

Rates of retention (% participants retained/participants recruited) and exclusion (% participants excluded/participants recruited) after one year of intervention were calculated. For participants excluded from the study, the duration of the intervention was computed. Participants who remained in the intervention include all participants not formally excluded from the study. The fidelity of intervention delivery was assessed by the elapsed time between cancer diagnosis and initial evaluation. Participants’ level of engagement was determined subjectively by the RDs after one year of intervention based on the ease of scheduling appointments and the global interest towards the intervention. Level of engagement was classified as low: participants who showed minimal involvement during visits (no questions asked, not interested in making dietary changes), multiple refusals and/or avoidance of meetings; moderate: participants who attended visits but were passively involved during visits and/or if multiple appointments were required to complete a follow-up (many postponed follow-ups); and high: participants who were actively involved during visits (many questions asked and demonstration of interest in nutritional counseling) and ease of scheduling/conducting follow-up visits. The participation rate of the intervention was computed for each participant by the number of follow-up visits completed/6 planned). Participants were grouped according to their level of participation, defined as low (0–1 visit); moderate (2–3 visits), and high (≥4 visits).

For each visit, attendance was defined as the number of participants who attended a visit divided by the total number of potential participants (i.e., who were still included in the study at each visit) and expressed as a percentage (% actual/ potential attendance). The completion rate of study measures at each visit (% performed/ planned) was also computed.

Based on other studies assessing the feasibility of an intervention for children and their parents, the following feasibility criteria were pre-defined:Eligibility: ≥70% of patients are eligible for the intervention [48,49].Recruitment: ≥50% of patients approached are recruited [50,51,52].Retention: ≥75% of participants are retained in the intervention after one year [49,51].Attendance: ≥70% of participants attend the planned visits (initial evaluation and follow-up visits) [50,51].Completion of study measures: acceptable if ≥70% of participants complete the measures and feasible if ≥85% of participants do so [48,51].

### 2.7. Assessment of Dietary Intakes

The 24H-R and 3-day food records were analyzed using the web application Nutrific^®^ developed by the Department of Food Science and Nutrition, Université Laval (https://nutrific.fsaa.ulaval.ca, accessed on 11 February 2022) based on the 2010 Canadian Nutrient File. Vitamin supplements and intravenous fluids were not considered in the analysis. Intakes of energy, fat, protein, dietary fiber, sodium, calcium, and vitamins C and D were calculated for each participant. Energy intake was reported as total calories (kcal) and per body weight (kcal/kg). Fat intake was calculated as gram per body weight (g/kg) and as the contribution of energy expressed in percent (% energy). Protein intake was computed on kg of body weight (g/kg), as a percentage of total energy (% energy), and as a percentage of the recommended dietary allowance (RDA) based on the Canadian dietary reference intakes (RDI) [53]. Intakes of dietary fiber and nutrients (sodium, calcium, vitamin C, and vitamin D) were adjusted for energy (per 1000 kcal) and presented as the percentages of the adequate intake (AI) for dietary fiber, of the tolerable upper intake level (UL) for sodium, and the RDA for calcium, and vitamins C and D [53]. The 24H-R completed at initial evaluation was used to describe the dietary intake before the intervention. For the post-intervention time point, we analyzed all the 24H-Rs that were completed (12 ± 2 months after the initial evaluation).

### 2.8. Data Analysis

Participants’ characteristics are described as mean ± standard deviation (SD), median, and range (min–max) for continuous variables {age at recruitment, z-score of BMI or W/L ratio, and serum 25(OH)D}. Categorial data including socio-demographic characteristics, cancer-related information, and nutritional and vitamin D status are presented as a percentage (%) of the total of participants. Descriptive statistics were used to report a feasibility assessment by presenting numbers and percentages for retention, abandon, engagement level, attendance, and completion of study measures. Continuous (duration of the intervention, time between diagnosis and initial evaluation) and ordinal variables {participation to follow-up visits (number of completed visits and participation rate)} are presented as mean ± SD, median, and range. Feasibility data and participants’ characteristics (socio-demographic, cancer-related information, and nutritional status) were compared between participants retained in the intervention and those who were excluded using Pearson’s Chi-square or Fisher’s exact tests (to assess the relationship between nominal variables) and Mann-Whitney tests (to investigate the differences between continuous variables). Dietary intakes are presented as mean, range (min–max), median, and interquartile range (25th to 75th percentiles). Participants’ characteristics and dietary intakes at initial evaluation were also compared according to participation level {low (0–1 visit), moderate (2–3 visits), or high (≥ 4 visits)} using Pearson’s Chi-square or Fisher’s exact tests and Kruskall-Wallis *H*-tests. Analyses were conducted to investigate the impact of the COVID-19 pandemic. Feasibility parameters were compared between participants who completed the intervention before the pandemic and those who were affected by it using Pearson’s Chi-square or Fisher’s exact tests and Mann-Whitney tests. Dietary intakes were compared between initial evaluation and post-intervention using Wilcoxon or paired sample *t*-tests. Analyses were performed using SPSS version 25.0 (IBM, Armonk, NY, USA). A *p*-value <0.05 was considered statistically significant.

## 3. Results

### 3.1. Recruitment and Description of the Cohort

During the study period, 147 patients were identified as potential participants for the VIE intervention according to the inclusion and exclusion criteria (Figure 1). Before reaching out to patients and their parents, approval from the healthcare team (attending oncologist and healthcare professionals) was required. Approval was not granted when a patient had serious medical complications or when the family or emotional context was considered too complex (*n* = 22, 15.0%). For 15 patients (10.2%), authorization from the medical team could not be obtained before the limit of the recruitment period (12 weeks after diagnosis). One patient died before the study could be presented. Therefore, a total of 109 patients were first identified to participate in the VIE study. Among them, consent forms were not presented to 3 families as they did not understand/read French and the consent form was only available in French at the time. We were also unable to contact 14 families before the end of the recruitment period. In the end, the study consent form was presented to 92 participants, which represents an eligibility rate of 62.6% (target for success ≥70%). Informed consent was obtained for 62 patients, corresponding to a 67.4% recruitment rate (target for success ≥50%).

The 62 participants that were recruited in the VIE study and their characteristics are described in Table 1. The distribution of participants by sex was almost equal (*n* = 32 boys; 51.6%). The mean age at recruitment was 8.5 years (range of 1.4 to 17.3) and proportions of preschoolers, children, and adolescents were similar. Most parents were married or common-law partners (79.2%), more than half had a university degree (56.4%), and more than one-third had an approximated gross family income of less than $70,000 (35.3%). Overall, 29 patients (46.8%) were diagnosed with leukemia and during the one-year intervention, 11 participants (17.7%) had a refractory response to cancer treatment or a relapse (non responders). At initial evaluation, based on BMI or W/L ratio, no participant was identified as malnourished, whereas 15 (24.2%) were classified as overweight or obese. The mean serum concentration of 25(OH)D was 59.8 nmol/L (SD: 24.4 nmol/L), and 69.2% (*n* = 36) of participants had sufficient vitamin D levels whereas 3 patients were classified as deficient (5.8%).

### 3.2. Feasibility of the Intervention

As illustrated in the flow diagram (Figure 1), the retention rate was 72.6% (target for success ≥75%) as 45 participants remained in the intervention (exclusion rate: 27.4%). Only one participant was never met by the RDs because of withdrawal from the study prior to the initial evaluation following healthcare providers’ recommendation. This participant was excluded from further analyses and subsequent analyses, therefore, there was a total of 61 participants. The reasons for exclusion from the nutritional intervention can be classified as medical and non-medical (Figure 1). For 6 participants, worsening of medical condition resulted in study withdrawal, including one death during the intervention. Study withdrawal following admission in palliative care was recommended by the healthcare providers for 5 participants (two of them died shortly after). For 10 participants, the reasons were non-medical and included the loss of interest in the study because of the time commitment it required (*n* = 3), the context related to the COVID-19 pandemic (*n* = 2), and failure of the intervention to meet expectations (*n* = 1). A total of 4 participants were lost during follow-up as they could not be reached, rendering follow-up visits impossible to schedule.

The duration of the intervention for the 16 participants who were excluded from the study ranged from 0 to 10.8 months (Table 2). Intervention duration was shorter in participants excluded because of study drop-out or loss during follow-up, compared to those excluded due to their medical conditions (median: 4.4 vs. 7.2 months; *p* = 0.09). For all 6 participants excluded for medical conditions, the cause was cancer related. One patient died suddenly from complications caused by cancer progression and the others were excluded after their transfer to palliative care due to early relapse after treatment (*n* = 1), tumor progression during treatment (*n* = 3), and treatment-refractory cancer (*n* = 1).

The nutritional intervention was initiated after a mean time of 13.2 weeks from diagnosis (range: 3.1–43.0 weeks) (Table 2). There was no difference in the median time to perform initial evaluation between participants retained in the intervention and those excluded. The engagement level in the intervention was established by the RDs for 55 participants. A majority of participants had a high level of engagement (56.4%). Overall, the mean number of follow-up visits completed was 3.2, representing a mean rate of 53.8% of planned visits. Participants excluded from the intervention, compared to those retained, were more likely to have a low level of engagement (50% vs. 8.9%; *p* < 0.01) and had a lower participation rate (median of 16.7% vs. 66.7%; *p* < 0.001). Participants with refractory disease or relapse were less likely to be retained in the intervention compared to treatment responders (45.5% vs. 80.0%; *p* = 0.03) (Figure 2). Nevertheless, non responders had a similar engagement level compared to participants who were responders to cancer treatment (data not shown). No other association between participants’ characteristics and intervention retention was found (data not shown).

Table 3 shows the attendance rate at the initial and follow-up visits. Overall, participants attended 258 out of the 362 planned visits, corresponding to a 71.3% attendance rate (target for success ≥70%). The attendance rate decreased over the course of the study, ranging from 83.9% (follow-up visit 1) to 48.9% (follow-up visit 6). The average attendance rate was 64.8% (95% CI: 54.1–75.4%). The attendance rate was not different between participants who completed the intervention before the pandemic and those who were affected by it (data not shown).

At initial evaluation, completion of the 24H-R was found more feasible than of the 3-day food record (Table 3): the low completion rate of this tool (37.7%) resulted in discontinuing its usage for follow-up visits. Measuring weight and height to calculate BMI and W/L were the most feasible anthropometric measures at initial evaluation followed by the MUAC, TSFT, SSFT, and WC. At follow-up visits, the 24H-R and BMI or W/L were the two study measures with the highest completion rates, ranging from 72.7% to 89.4% and from 86.4% to 97.1%, respectively. In contrast, WC and SSFT measures had the lowest completion rates (ranging from 4.4% and 31.8%). There was a relationship between the completion of anthropometric measurements (WC, MUAC, TSFT, and SSFT) and age group (children vs. adolescents). At initial evaluation, all four parameters were completed by 55.6% of adolescents (*n* = 10/18) but only by 18.6% of children (*n* = 8/35), which was statistically significant (*p* < 0.01). A supplementary analysis was conducted to assess the relationship between attendance, study measure completion rates, and retention in the intervention (Appendix A Appendix A). Overall, no pattern of associations between these variables was observed. We also assessed the relationship between feasibility data (engagement level, participation to follow-up visits, attendance, and completion rates), participants’ age category (preschoolers vs. children vs. adolescents), and cancer diagnosis (leukemia vs. lymphoma vs. sarcoma vs. other). No consistent trend was identified between these variables (data not shown).

Participants were grouped according to their level of participation, defined as low (0–1 visit); moderate (2–3 visits); and high (≥4 visits) (Table 4). Half of the participants were considered to have high participation (*n* = 31, 50.8%), whereas 31.1% and 18.0% were classified as having moderate and low level, respectively. There was a tendency towards a relationship between age category and participation level as 60.5% of children were classified with high participation compared to 27.5% of adolescents (*p* = 0.06). Participants with a high level of engagement were more likely to have attended ≥4 follow-up visits (high participation level) compared to those with moderate and low engagement levels (*p* = 0.001). No difference in participation level was found according to participants’ socio-demographic profile, cancer-related characteristics, and nutritional status.

Dietary intakes, as assessed at the initial evaluation, were compared between participants according to their participation level (Table 5). Overall, there was no difference in energy and nutrient intake amongst groups, except for proteins, as those with higher participation levels also had a higher percentage of their total energy provided by proteins.

Changes in dietary intake was assessed in 21 participants (Table 6). After a mean time of 12 months (SD: 1.1 month), participants had a lower intake of sodium {median (IQR): 1065.3 (950.7–1190.6) mg/1000 kcal} compared to the initial evaluation {1542.5 (1078.4–1665.6) mg/1000 kcal; *p* = 0.03}. We also found a trend for a higher intake in vitamin C (median: 366.6% vs. 98.1%; *p* = 0.08).

## 4. Discussion

The results of this study support the feasibility of conducting a one-year nutritional intervention with a bi-monthly follow-up that promotes healthy eating behaviors and which begins early after the diagnosis of pediatric cancer. Both the intervention recruitment and attendance rates were above the pre-determined targets for success. Although the eligibility (62.5%) and retention (72.6%) rates were slightly below the established targets of ≥70% and ≥75%, we believe that the difference is too modest to conclude that the intervention is not feasible. After one year, 72.6% of participants were still active in the study, and two-thirds of the scheduled follow-up visits were completed. The BMI or W/L ratio and 24H-R were easier study measures to collect compared to WC, SSFT, and 3-day food records. We also found that participants who did not respond to cancer treatment (had relapsed or refractory disease) were most likely to be excluded from the intervention. Moreover, patients retained in the intervention and those with high participation level (≥4 follow-up visits) were more likely to have a high level of engagement. In addition, at initial evaluation, nearly 25% of participants were overweight or obese and 30.8% had serum vitamin D levels below 50 nmol/L indicating a risk of insufficiency for bone health [47]. These results support not only the feasibility of the intervention but also its relevance in a pediatric oncology setting.

Malnutrition (undernutrition and overnutrition) at pediatric cancer diagnosis and during treatment has been associated with worse outcomes, more treatment-related toxicities, and poor quality of life [17,54,55,56,57]. Interventions that promote the adoption or the maintenance of healthy lifestyle habits during treatment may improve patients’ nutritional status and dietary intakes. Therefore, in the short term, they could lead to a better prognosis and quality of life, while in the long term, they could help prevent the development of health complications. However, the studies assessing the impact of such nutritional intervention in pediatric cancer have been conducted towards the end of the treatment trajectory, i.e., during the maintenance phase or after therapy [30,32,58]. While limited in number and to children with ALL, these studies have shown little impact on weight gain, even if dietary intakes were improved [32,33,58]. It is possible that interventions starting earlier after diagnosis would be more appropriate than those initiated later during treatment or targeting survivors. Here, we describe the feasibility of a nutritional intervention initiated after a median time of 11.9 weeks following diagnosis of pediatric cancer.

From the time of cancer diagnosis, children and adolescents face many challenges that can be detrimental to the development and maintenance of a long-term healthy lifestyle. Preschool age is a critical period in the development of eating habits [41,59]. As about half of all childhood cancer diagnoses (0–14 years) occur before the age of 5 [1], and given the impact of antineoplastic treatments on nutritional outcomes, this may negatively impact children’s feeding practices [41,60]. In addition, parental attitudes are often modified following the diagnosis of childhood cancer as they tend to be more permissive towards dietary choices and use food as a reward [30,41,61]. Changes in eating habits, sedentary behavior, and weight gain acquired by children and adolescents early in the cancer treatment trajectory may persist after therapy [30,62,63]. Thus, researchers and parents expressed the need for interventions to begin early after diagnosis in order to support families with feeding practices that may prevent negative long-term eating habits [30,41,61]. However, the timing of initiating such nutritional intervention in different types of pediatric cancer has been poorly explored. Our study did not reveal differences in the time to initiate the intervention from cancer diagnosis between retained and excluded participants. Moreover, the level of participation in the intervention did not differ according to the type of cancer diagnosis. These results support that it is feasible to start a nutritional intervention 4 to 12 weeks after pediatric cancer diagnosis regardless of its type.

It has been proposed that the emotional impact of childhood cancer diagnosis and the acute treatment phase have been identified as barriers to implementing such an early intervention [30,64,65]. Accordingly, in our study, the worsening of the medical condition was one of the main reasons for exclusion from the intervention. However, participants identified as non responders to cancer treatment (had relapsed or refractory disease) were not all excluded from the study, as 5 out of 11 remained in the intervention (including 1 deceased). Additionally, the 11 participants who were classified as non responders to cancer treatment had similar engagement and participation levels compared to the other patients. This suggests that the intervention can still be feasible for these participants and that they should not be excluded outright because of an unfavorable medical condition.

Participants excluded from the study for non-medical reasons had a short duration of the active intervention (median: 4.4 months) and a low number of follow-up visits (median: 1 visit; range: 0–3). This indicates that these participants recognized early on that the intervention was not suitable for them and either dropped out from the study or avoided the appointments (lost during follow-up). Two participants dropped out from the intervention due to the COVID-19 pandemic circumstances: the follow-up visits planned at the outpatient clinic were replaced by telephone monitoring.

The attendance rate decreased over time during the one-year intervention, but the COVID-19 pandemic did have a significant impact. In a pilot study designed for parents, six families were recruited to test the feasibility and acceptability of an intervention which consisted of six group information sessions (1 per week) related to overweight and obesity in pediatric cancer survivors [30]. Most of the parents attended at least 75% of the program, but there was no additional information on the attendance rate for each session; although parents expressed uncertainty about their availability to participate in all six sessions [30]. The necessity to attend multiple sessions has been identified as a barrier to participation among parents of pediatric cancer survivors. In a randomized nutritional counseling intervention including 12 children with ALL, the one-year intervention comprising of 12 monthly one-on-one follow-up visits was successfully delivered [66]. However, no specific data on feasibility such as rates of attendance or completion of study measures were documented or reported [66]. Feasibility evaluation, refers to answering the question “can this study be done?” [34,67], has been poorly described in other nutritional intervention studies in pediatric oncology. To the best of our knowledge, our study is the first to address this issue for all types of childhood cancer.

In our study, a lower proportion of adolescents was classified with high participation level (≥4 follow-up visits) compared to children. Adolescents with cancer have specific concerns and needs that differ from their younger peers [68]. Nutritional interventions through individual counseling are less effective in adolescents than school-based programs [69]. It is possible that the design of our intervention based on one-on-one visits was less appropriate to stimulate optimal participation in adolescents compared with children and their parents.

Assessing the feasibility of anthropometric measures for nutritional evaluation revealed that WC had the lowest completion rates. Among arm anthropometry parameters, MUAC had the highest completion rate and SSFT the lowest. Logistical challenges may explain this result. While measures were performed in the outpatient clinic, a private room was not always available, making it difficult to take certain measures. Furthermore, some patients were bedridden and wearing only a hospital gown. Therefore, sometimes the measurements that require the patient to stand up to access the body measurement site were not performed to respect the patient’s condition and/or privacy. Age was also a determining factor in the completion of anthropometric measurements. Taking anthropometric measurements was found to be more difficult in infants and children because they were often fearful of medical manipulations, especially those involving the caliper used to assess the TSFT and SSFT.

In the context of our study, the 24H-R was the best tool to collect dietary intakes. Completion of the 3-day food journals by participants was not successful at the initial visit, thus the tool was later abandoned for follow-up visits. We can speculate that the burden of completing the tool by families was too much early after diagnosis. Next, we did not uncover straightforward relationships between feasibility parameters and participants’ quality of diet at the initial evaluation. Only the greater protein intakes of patients with high participation level suggest better diet quality for this group. However, these data remain insufficient to draw clear conclusions and further studies are needed to examine a possible association between diet quality and participation in a nutritional intervention in the context of pediatric oncology. Our preliminary assessment of the impact of the intervention suggests that participants slightly improved their diet, with regard to sodium intake. However, it is possible that this is related to other factors, such as fewer side effects of cancer treatment, which could lead to a better diet. Further studies are needed to assess the impact of such intervention on the quality of diet and food intake.

Based on our experience and the results of this study, a number of actions can be taken to improve participation in a nutritional intervention early after a pediatric cancer diagnosis. Confirming the follow-up visits in advance (by phone or email) could help increase attendance rates and participation. Additionally, since adolescents tended to have lower participation levels than their younger peers, we believe that adapting the nutritional intervention to teenagers could improve their involvement and resulting outcomes.

## 5. Conclusions

Our study supports that a one-year nutritional intervention is feasible early after any pediatric cancer diagnosis. Given the potential benefit of promoting healthy lifestyle habits during the acute phase of treatment and in long-term cancer survivors, early nutritional interventions should be considered. Further studies are needed to evaluate the impact of nutritional intervention on eating habits or health outcomes during and after treatment.

## Figures and Tables

**Figure 1 nutrients-14-01024-f001:**
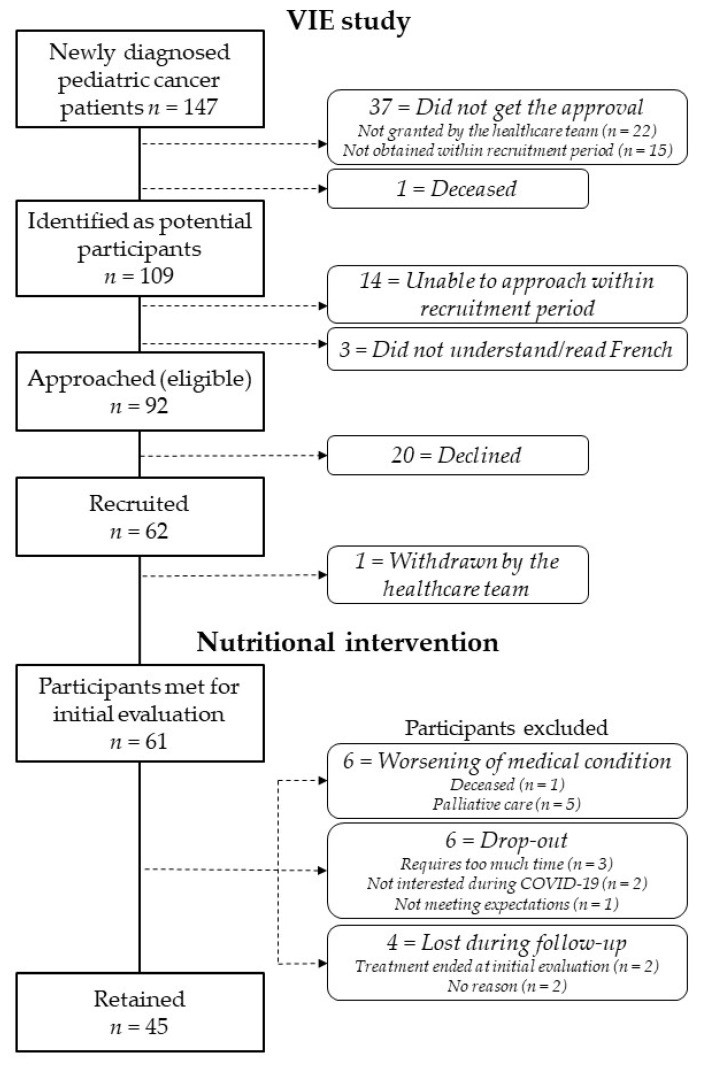
Flow diagram of participants recruited for the nutritional intervention.

**Figure 2 nutrients-14-01024-f002:**
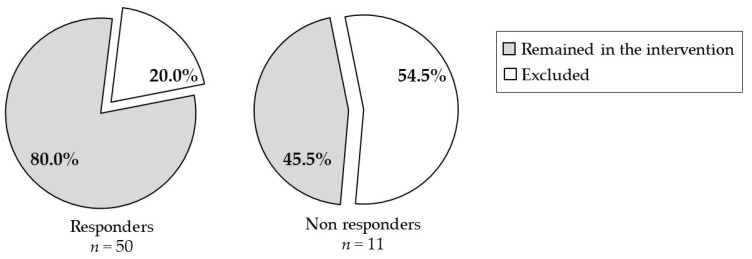
Distribution of participants’ retention in the intervention based on their cancer evolution. The percentage of patients retained in the intervention and those excluded is illustrated according to participants’ cancer evolution (responders vs. non responders). Data were compared using Fisher’s exact test.

**Table 1 nutrients-14-01024-t001:** Characteristics of participants.

	Participants
Sex, *n* (%)	*n* = 62
Male	32 (51.6)
Female	30 (48.4)
Age at recruitment, year	*n* = 62
Mean ± SD	8.5 ± 5.1
Median (min–max)	7.4 (1.4–17.3)
Age category, *n* (%)	*n* = 62
Preschoolers (<5 y.o.)	22 (35.5)
Children (5 to 12 y.o.)	22 (35.5)
Adolescents (≥13 y.o.)	18 (29.0)
Marital status, *n* (%)	*n* = 53
Married/common-law partners	42 (79.2)
Separated/divorced/widower	11 (20.8)
Parental education level, *n* (%)	*n* = 55
Unfinished high school	2 (3.6)
High school	13 (23.6)
College	9 (16.4)
University	31 (56.4)
Approximated gross family income, *n* (%)	*n* = 51
<$29,999	4 (7.8)
$30,000–$69,999	12 (23.5)
$70,000–$109,999	17 (33.3)
$110,000–$150,000	11 (21.6)
>$150,000	7 (13.7)
Cancer diagnosis, *n* (%)	*n* = 62
Leukemia ^1^	29 (46.8)
Lymphoma ^2^	12 (19.4)
Sarcoma ^3^	10 (16.1)
Other ^4^	11 (17.7)
Cancer evolution, *n* (%)	*n* = 62
Responders	51 (82.3)
Non responders ^5^	11 (17.7)
BMI or W/L, z-score	*n* = 61
Mean ± SD	0.3 ± 1.1
Median (min–max)	0.1 (−1.9–2.8)
Nutritional status, *n* (%)	*n* = 61
Normal	47 (75.8)
Overweight/Obese	15 (24.2)
Serum 25(OH)D, nmol/L	*n* = 52
Mean ± SD	59.8 ± 24.4
Median (min–max)	58.2 (22.0–168.0)
Vitamin D status, *n* (%)	*n* = 52
Sufficiency	36 (69.2)
Insufficiency	13 (25.0)
Deficiency	3 (5.8)

SD: standard deviation; y.o.: years old; BMI: body mass index; W/L: weight-for-length ratio; 25(OH)D: 25-hydroxyvitamin D. ^1^ Leukemia diagnosis includes acute lymphoblastic leukemia (*n* = 26) and acute myeloid leukemia (*n* = 3). ^2^ Lymphoma diagnosis includes Hodgkin’s lymphoma (*n* = 2), Burkitt’s lymphoma (*n* = 6), anaplastic lymphoma (*n* = 2), diffuse large B-cell lymphoma (*n* = 1), and lymphoblastic lymphoma (*n* = 1). ^3^ Sarcoma diagnosis includes osteosarcoma (*n* = 2), rhabdomyosarcoma (*n* = 3), Ewing’s sarcoma (*n* = 4), and undifferentiated sarcoma (*n* = 1). ^4^ Other diagnosis includes Wilm’s tumor (*n* = 3), thymoma (*n* = 1), germinoma (*n* = 2), medulloblastoma (*n* = 1), hepatic rhabdoid tumor (*n* = 1), neuroblastoma (*n* = 2), and hepatoblastoma (*n* = 1).^5^ Non responders designate patients who had a relapse or refractory disease.

**Table 2 nutrients-14-01024-t002:** Feasibility data and comparison between participants retained and excluded.

	Participants	Retained	Excluded	
	*n* = 61	*n* = 45	*n* = 16	*p*-Value
Duration of the intervention, months				-
Mean ± SD	N/A	N/A	4.6 ± 3.3	
Median (min–max)	N/A	N/A	4.9 (0–10.8)	
Time between diagnosis and initial evaluation, weeks				0.34
Mean ± SD	13.2 ± 7.1	13.1 ± 7.7	13.7 ± 4.9	
Median (min–max)	11.9 (3.1–43.0)	11.7 (3.1–43.0)	12.4 (5.86–22.0)	
Engagement level ^1^, *n* (%)	*n* = 55	*n* = 45	*n* = 10	<0.01
Low	9 (16.4)	4 (8.9)	5 (50.0)	
Moderate	15 (27.3)	15 (33.3)	0	
High	31 (56.4)	26 (57.8)	5 (50.0)	
Participation in follow-up visits				
Number of follow-up visits completed, *n*				<0.001
Mean ± SD	3.2 ± 1.7	4.0 ± 1.1	1.2 ± 1.2	
Median (min–max)	4.0 (0–6.0)	4.0 (2.0–6.0)	1.0 (0–4.0)	
Rate ^2^, %				<0.001
Mean ± SD	53.8 ± 27.6	65.9 ± 18.5	19.8 ± 19.5	
Median (min–max)	66.7 (0–100)	66.7 (33.3–100)	16.7 (0–66.7)	

The differences in feasibility parameters were assessed according to participants’ retention in the intervention using Mann-Whitney *U*-tests or Pearson Chi-square test. SD: standard deviation. ^1^ Level of engagement was determined subjectively by the RDs after one year of intervention based on the ease of scheduling follow-up appointments and the global interest towards the intervention. ^2^ Percentage of follow-up visits completed out of the six planned.

**Table 3 nutrients-14-01024-t003:** Attendance and completion rates of study measures at visits.

	Initial Evaluation	2-Month Follow-Up	4-Month Follow-Up	6-Month Follow-up	8-Month Follow-Up	10-Month Follow-Up	12-Month Follow-Up	All Visits
**Attendance**								
Actual/ potential, *n*	61/62	47/56	38/55	35/51	32/47	23/46	22/45	258/362
Rate (%)	(98.4)	(83.9)	(69.1)	(68.6)	(68.0)	(50.0)	(48.9)	(71.3)
**Completion rates, *n* (%)**								
Study measures	(*n* = 61)	(*n* = 47)	(*n* = 38)	(*n* = 35)	(*n* = 32)	(*n* = 23)	(*n* = 22)	Mean (95% CI)
3-day food record	23 (37.7)	-	-	-	-	-	-	-
Blood sample	55 (90.2)	-	-	-	-	-	-	-
24H-R	53 (86.9)	42 (89.4)	31 (81.6)	28 (80.0)	27 (84.4)	20 (87.0)	16 (72.7)	83.1 (78.6–87.6)
BMI or W/L	61 (100)	45 (95.7)	35 (92.1)	34 (97.1)	30 (93.8)	22 (95.7)	19 (86.4)	94.4 (91.0–97.8)
WC	20 (32.8)	7 (14.9)	3 (7.9)	3 (8.6)	3 (9.4)	1 (4.4)	7 (31.8)	15.7 (6.3–25.1)
MUAC	37 (60.7)	19 (40.4)	14 (36.8)	8 (22.9)	7 (21.9)	5 (21.7)	9 (40.9)	35.0 (23.6–46.4)
TSFT	29 (47.5)	18 (38.3)	11 (29.0)	6 (17.1)	7 (21.9)	4 (17.4)	7 (31.8)	29.0 (20.0–38.0)
SSFT	23 (37.7)	14 (29.8)	8 (21.1)	4 (11.4)	6 (18.8)	1 (4.4)	5 (22.7)	20.8 (11.9–29.7)

Attendance was defined as the number of participants who attended a visit divided by the total number of potential participants. Completion rates of study measures (24H-R, BMI, WC, MUAC, TSFT, and SSFT) were calculated based on participants who attended each visit and expressed as a percentage. CI: confidence intervals; 24H-R: 24-h recall; BMI: body-mass-index; W/L: weight-for-length ratio; WC: waist-circumference; MUAC: mid-upper arm circumference; TSFT: triceps skinfold thickness; SSFT: subscapular skinfold thickness.

**Table 4 nutrients-14-01024-t004:** Participants’ characteristics according to their participation level.

	Participation Level			
	Low	Moderate	High	
	(0–1 visit)	(2–3 Visits)	(≥4 Visits)	
	*n* = 11	*n* = 19	*n* = 31	*p*-Value
Sex, *n* (%)				0.40
Male	5 (45.5)	8 (42.1)	19 (61.3)	
Female	6 (54.5)	11 (57.9)	12 (38.7)	
Age at recruitment, year				0.36
Mean ± SD	9.7 ± 5.7	9.4 ± 5.2	7.4 ± 4.9	
Median (min–max)	8.9 (2.4–17.1)	8.8 (1.9–16.3)	5.2 (1.3–17.1)	
Age category, *n* (%)				0.06
Children (<13 y.o.)	6 (54.5)	11 (57.9)	26 (83.9)	
Adolescents (≥13 y.o.)	5 (45.5)	8 (42.1)	5 (16.1)	
Marital status, *n* (%)	*n* = 10	*n* = 16	*n* = 27	0.74
Married/common-law partners	9 (90.0)	12 (75.0)	21 (77.8)	
Separated/divorced/widower	1 (9.1)	4 (25.0)	6 (22.2)	
Parental education level, *n* (%)	*n* = 10	*n* = 16	*n* = 29	0.86
Unfinished high school/high school	4 (40.0)	4 (25.0)	7 (24.1)	
College	1 (10.0)	2 (12.5)	6 (20.7)	
University	5 (50.0)	10 (62.5)	16 (55.2)	
Mean gross family revenue, *n* (%)	*n* = 9	*n* = 14	*n* = 28	0.08
<$29,999	1 (11.1)	1 (7.1)	2 (7.1)	
$30,000–$69,999	1 (11.1)	2 (14.3)	9 (32.1)	
$70,000–$109,999	7 (77.8)	3 (21.4)	7 (25.0)	
$110,000–$150,000	1 (11.1)	4 (28.6)	7 (25.0)	
>$150,000	0	4 (28.6)	3 (10.7)	
Diagnosis, *n* (%)				0.31
Leukemia	3 (27.3)	9 (47.4)	16 (51.6)	
Lymphoma	2 (18.2)	4 (21.1)	6 (19.4)	
Sarcoma	2 (18.2)	5 (26.3)	3 (9.7)	
Other	4 (36.4)	1 (5.3)	6 (19.4)	
BMI or W/L, z-score				1.00
Mean ± SD	0.3 ± 1.1	0.3 ± 1.3	0.3 ± 1.1	
Median (min–max)	−0.1 (−1.0–2.5)	−0.1 (−1.9–2.8)	0.1 (−1.7–2.3)	
Nutritional status, *n* (%)				0.92
Normal	9 (81.8)	14 (73.7)	25 (80.6)	
Overweight/Obese	2 (18.2)	5 (26.3)	7 (19.4)	
Cancer evolution, *n* (%)				0.68
Responders	8 (72.7)	16 (84.2)	26 (83.9)	
Non responders ^1^	3 (27.3)	3 (15.8)	5 (16.1)	
Engagement level ^2^, *n* (%)	*n* = 7	*n* = 17	*n* = 31	0.001
Low	5 (71.4)	3 (17.6)	1 (3.2)	
Moderate	0	7 (41.2)	8 (25.8)	
High	2 (28.6)	7 (41.2)	22 (71.0)	

The participation level was determined for each participant based on the number of follow-up visits attended and defined as low (0–1 visit); moderate (2–3 visits); and high (≥4 visits). The relationships between participation level and characteristics (nominal variables) were evaluated using Pearson’s Chi-square or Fisher’s exact tests. The differences in continuous variables of participants according to their participation level were assessed using Kruskal-Wallis *H*-test. SD: standard deviation; y.o.: years old; BMI: body mass index; W/L: weight-for-length ratio. Leukemia diagnosis includes acute lymphoblastic leukemia (*n* = 25) and acute myeloid leukemia (*n* = 3). Lymphoma diagnosis includes Hodgkin’s lymphoma (*n* = 2), Burkitt’s lymphoma (*n* = 6), anaplastic lymphoma (*n* = 2), diffuse large B-cell lymphoma (*n* = 1), and lymphoblastic lymphoma (*n* = 1). Sarcoma diagnosis includes osteosarcoma (*n* = 2), rhabdomyosarcoma (*n* = 3), Ewing’s sarcoma (*n* = 4), and undifferentiated sarcoma (*n* = 1). Other diagnosis includes Wilm’s tumor (*n* = 3), thymoma (*n* = 1), germinoma (*n* = 2), medulloblastoma (*n* = 1), hepatic rhabdoid tumor (*n* = 1), neuroblastoma (*n* = 2), and hepatoblastoma (*n* = 1). ^1^ Non responders designate patients who had a relapse or refractory disease. ^2^ Level of engagement was determined subjectively by the RDs after one year of intervention based on the ease of scheduling appointments and the global interest towards the intervention.

**Table 5 nutrients-14-01024-t005:** Dietary intakes at initial evaluation according to participation level.

		Participation Level		
	Low	Moderate	High	
	(0–1 Visit)	(2–3 Visits)	(≥4 Visits)	
	*n* = 9	*n* = 19	*n* = 25	*p*-Value
Energy				
Kcal				
Mean (min–max)	1976 (767–3040)	1418 (224–2698)	1702 (9–3751)	0.22
Median (IQR)	2077 (1696–2522)	1452 (842–2058)	1508 (1075–2149)	
Kcal/kg				
Mean (min–max)	77.3 (19.5–194.0)	51.4 (4.9–115.1)	64.8 (0.4–159.5)	0.48
Median (IQR)	48.8 (39.5–112.3)	51.9 (27.9–74.4)	50.5 (35.2–86.1)	
Fat				
g/kg				
Mean (min–max)	3.4 (0.9–8.6)	2.0 (0.2–4.9)	2.7 (0–8.7)	0.40
Median (IQR)	2.1 (1.4–4.8)	2.0 (0.8–2.7)	2.2 (1.0–3.4)	
% Energy				
Mean (min–max)	38.4 (27.5–49.4)	35.2 (23.6–53.3)	35.4 (2.3–56.1)	0.50
Median (IQR)	38.0 (36.9–39.9)	35.1 (29.5–40.0)	37.4 (31.0–39.5)	
Protein				
g/kg				
Mean (min–max)	2.7 (0.6–7.8)	2.1 (0.1–4.2)	2.8 (0–7.5)	0.66
Median (IQR)	1.6 (1.2–3.7)	2.5 (1.0–3.1)	2.2 (1.4–4.1)	
% Energy				
Mean (min–max)	13.5 (11.4–16.1)	15.9 (5.0–20.3)	16.7 (1.0–22.9)	0.02
Median (IQR)	13.3 (12.8–14.2)	16.0 (13.9–19.1)	18.1 (13.7–19.3)	
% RDA				
Mean (min–max)	273.1 (67.0–741.9)	218.0 (7.2–440.8)	289.0 (0–712.1)	0.69
Median (IQR)	184.1 (144.4–354.6)	267.1 (107.1–325.0)	232.4 (146.8–404.7)	
Dietary fiber				
Energy-adjusted (g/1000 kcal)				
Mean (min–max)	9.8 (3.1201321.3)	8.2 (3.4–15.0)	7.6 (0–18.0)	0.50
Median (IQR)	9.2 (6.6–10.8)	7.7 (6.2–10.2)	7.4 (5.3–9.4)	
% AI				
Mean (min–max)	88.9 (12.3–283.0)	45.1 (5.9–92.8)	43.9 (0–107.0)	0.05
Median (IQR)	73.6 (58.8–83.7)	50.5 (19.8–63.6)	41.5 (32.5–56.7)	
Sodium				
Energy-adjusted (mg/1000 kcal)				
Mean (min–max)	1305.7 (423.2–1976.1)	1373.8 (551.5–2781.3)	1609.4 (29.9–7390.6)	0.81
Median (IQR)	1538.4 (954.7–1695.1)	1273.7 (1031.7–1665.3)	1484.3 (1118.7–1591.1)	
% UL				
Mean (min–max)	129.4 (63.3–224.0)	97.9 (12.0–240.8)	133.0 (0–313.1)	0.20
Median (IQR)	107.9 (71.2–178.5)	79.4 (42.9–143.8)	123.7 (80.2–165.5)	
Calcium				
Energy-adjusted (mg/1000 kcal)				
Mean (min–max)	464.5 (129.6–1016.6)	657.5 (65.1–1392.3)	619.1 (44.9–1175.7)	0.34
Median (IQR)	449.7 (285.1–536.5)	637.7 (466.3–792.4)	665.8 (276.1–875.5)	
% RDA				
Mean (min–max)	101.1 (7.7–260.4)	92.3 (2.5–224.6)	103.0 (0–334.3)	0.99
Median (IQR)	71.9 (58.7–124.8)	93.2 (46.6–120.8)	71.9 (40.0–177.2)	
Vitamin C				
Energy-adjusted (mg/1000 kcal)				
Mean (min–max)	59.2 (0–132.9)	46.6 (1.9–161.4)	81.3 (0.3–390.5)	0.56
Median (IQR)	68.3 (22.8–87.3)	29.2 (14.0–64.9)	58.2 (25.4–99.0)	
% RDA				
Mean (min–max)	329.5 (0.1–772.6)	251.0 (4.9–1379.9)	414.0 (0.9–2132.1)	0.52
Median (IQR)	275.4 (154.1–403.5)	101.7 (30.9–289.8)	220.5 (87.8–508.0)	
Vitamin D				
Energy-adjusted (μg/1000 kcal)				
Mean (min–max)	2.6 (0.0–8.8)	3.5 (0–10.6)	3.9 (0–13.3)	0.22
Median (IQR)	2.0 (1.6–2.5)	2.8 (1.1–4.3)	4.1 (1.6–4.9)	
% RDA				
Mean (min–max)	33.4 (0–105.5)	32.8 (0–79.8)	38.9 (0–101.0)	0.59
Median (IQR)	31.2 (8.0–39.5)	22.6 (11.7–48.9)	38.7 (22.3–55.3)	

Dietary intakes were evaluated in participants at initial evaluation using a 24-h recall. Differences in dietary intakes according to participation level were assessed using Kruskal-Wallis *H*-tests. IQR: interquartile range (25th to 75th percentiles); RDA: Recommended Dietary Allowance; AI: Adequate intake; UL: Tolerable upper intake level.

**Table 6 nutrients-14-01024-t006:** Comparison of dietary intake at initial evaluation and post-intervention.

	*n* = 21 Pairs		Difference ^1^	
	Initial Evaluation	Post-Intervention	Mean (95% CI)	*p*-Value
Energy				
Kcal				
Mean (min–max)	1584 (9–3751)	1822 (882–3386)	299 (–236–833)	0.25
Median (IQR)	1479 (1075–1900)	1727 (1362–2071)		
Kcal/kg				
Mean (min–max)	61.4 (0.4–159.5)	60.9 (17.9–120.4)	−0.4 (−17.2–16.4)	0.79
Median (IQR)	51.9 (38.3–78.4)	61.9 (41.0–78.0)		
Fat				
g/kg				
Mean (min–max)	2.7 (0–8.7)	2.3 (0.6–5.7)	−0.5 (−1.5–0.5)	0.73
Median (IQR)	2.2 (1.7–3.4)	2.0 (1.3–3.1)		
% Energy				
Mean (min–max)	37.1 (2.3–56.1)	33.2 (17.1–52.1)	−3.8 (−9.9–2.2)	0.20
Median (IQR)	38.5 (33.6–39.5)	28.7 (26.1–44.3)		
Protein				
g/kg				
Mean (min–max)	2.8 (0–7.5)	2.7 (0.4–5.7)	−0.1 (−1.0–0.7)	0.92
Median (IQR)	2.6 (1.5–3.3)	2.6 (1.8–3.5)		
% Energy				
Mean (min–max)	16.8 (1.0–22.3)	17.4 (9.3–42.5)	0.6 (−3.4–4.6)	0.87
Median (IQR)	18.9 (15.3–19.4)	16.6 (14.2–18.9)		
% RDA				
Mean (min–max)	289.0 (0–712.1)	281.9 (49.4–600.0)	−7.1 (−89.7–75.6)	0.86
Median (IQR)	268.0 (165.5–346.6)	282.1 (187.4–363.2)		
Dietary fiber				
Energy–adjusted (g/1000 kcal)				
Mean (min–max)	7.9 (0–18.0)	7.2 (1.9–15.1)	−0.7 (−2.9–1.5)	0.51
Median (IQR)	7.7 (5.9–9.9)	7.4 (5.7–8.3)		
% AI				
Mean (min–max)	44.5 (0–92.8)	48.1 (6.6–91.2)	3.5 (−11.1–18.2)	0.62
Median (IQR)	42.3 (32.5–57.5)	47.9 (41.1–53.2)		
Sodium				
Energy–adjusted (mg/1000 kcal)				
Mean (min–max)	1612.5 (29.9–7390.6)	1078.4 (727.7–1403.6)	−564.1 (−1193.1–64.9)	0.03
Median (IQR)	1542.5 (1078.4–1665.6)	1065.3 (950.7–1190.6)		
% UL				
Mean (min–max)	132.4 (0–313.1)	100.1 (59.1–206.6)	−32.3 (−81.9–17.4)	0.19
Median (IQR)	109.8 (74.9–165.5)	93.5 (68.2–114.8)		
Calcium				
Energy–adjusted (mg/1000 kcal)				
Mean (min–max)	614.9 (44.9–969.1)	581.2 (80.0–1520.2)	−33.7 (−181.6–114.3)	0.64
Median (IQR)	637.7 (549.7–773.5)	465.7 (349.2–723.7)		
% RDA				
Mean (min–max)	102.8 (0–334.3)	96.3 (20.5–216.4)	−6.5 (−43.8–30.8)	0.79
Median (IQR)	93.2 (64.2–109.8)	76.1 (45.7–140.6)		
Vitamin C				
Energy–adjusted (mg/1000 kcal)				
Mean (min–max)	70.8 (0.3–390.5)	86.8 (4.5–268.8)	15.9 (−35.0–66.9)	0.36
Median (IQR)	30.1 (16.4–63.5)	63.2 (33.6–111.3)		
% RDA				
Mean (min–max)	292.1 (0.9–2132.1)	436.0 (16.0–1307.7)	143.9 (−80.8–368.7)	0.08
Median (IQR)	98.1 (62.2–317.7)	366.6 (149.5–679.9)		
Vitamin D				
Energy–adjusted (μg/1000 kcal)				
Mean (min–max)	3.7 (0–13.3)	4.5 (0.3–33.1)	0.7 (−2.7–4.2)	1.00
Median (IQR)	4.0 (1.8–4.9)	3.0 (0.9–5.1)		
% RDA				
Mean (min–max)	36.4 (0–80.4)	47.9 (3.2–300.4)	3.5 (−11.1–18.2)	0.95
Median (IQR)	38.7 (20.3–53.0)	35.3 (8.7–58.1)		

Dietary intakes were evaluated in 21 participants at initial evaluation and post-intervention using 24-h recalls. Differences in mean dietary intakes before and after the intervention were compared using Wilcoxon or paired-samples *t*-tests. IQR: interquartile range (25th to 75th percentiles); RDA: Recommended Dietary Allowance; AI: Adequate intake; UL: Tolerable upper intake level.^1^ The mean difference in dietary intake of the group was calculated as dietary intake post-intervention-dietary intake at initial evaluation.

## Data Availability

The datasets used and/or analyzed are available from the corresponding author on reasonable request.

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
