# Peer review of "Early Nutritional Intervention to Promote Healthy Eating Habits in Pediatric Oncology: A Feasibility Study"

_nutrients, 2022, doi:10.3390/nu14051024_

Round 1

Reviewer 1 Report

Thank you for giving me the opportunity to review this paper. The authors are to be applauded by there effort to develop an intervention, aiming not only on undernutrition, but also on over nutrition

I do have some major  concerns.

When doing a feasibility study, it has to be decided beforehand when an intervention is deemed feasible, and when not.  I did not find this in the material and methods section. The conclusion at the start of the discussion that the results support the feasibility of the intervention therefore are incorrect in my opinion.

I understand a power analysis is not needed in this kind of studies, but why was a number of 62 patients recruited? Why not more, or less?

Feasibility was defined as % participants who completed the study etc. however, it was not registered where the dietician visits let to: did they attend the meetings for the sake of the study but did nothing with the info (what you might conclude from the lack of difference with respect to intake (besides protein) in relation to participation level

With 62 patients participating in the study, you might expect some data on its effectiveness. However, these data are not mentioned in the manuscript

Considering the above mentioned points, I find the manuscript too long: many data are given on items related to whether or not a patient participated for the whole study period, and on which items were more / less feasible to be measured. However, these data are not all new, and similar to outcomes in many other feasibility studies.

Reviewer 2 Report

The manuscript by Bélanger et al assesses the feasibility of applying a nutritional intervention already early after cancer diagnosis in pediatric cancer patients. The data shows that such intervention seems feasible across all age-groups and tumor types, although still several questions remain:

  • The authors indicate on line 75/76 that they “propose that a nutritional intervention that begins shortly after diagnosis and involves patient’s family is feasible”. In its current form it is however lacking how parents and family were involved in the nutritional intervention and as such this claim could not be made.
  • Was there a minimum age for recruitment?
  • What was the dietary composition that the participants received? This is unclear from the nutritional intervention part described in the material and methods. What was the advice for a more healthy eating pattern given by the RDs, or was this study simply for monitoring what patients were eating normally?
  • Was nutrient intake only assessed by quantifications from the 24H-R and 3-day food records or also measured in e.g. serum, and if so what method was used?
  • Many parents of pediatric oncology patients provide supplements to their children. Is it noted to how many parents (or percentage of participants) of the current study this applies to?
  • What was the inclusion rate? Did most children and parents that were asked to join indeed participate and if not for what reason were they not recruited?
  • The 6 participants that were excluded due to the medical conditions, were those treatment, cancer type or intervention related worsening?
  • With respect to the completion rates of the study. How can it be that 45 participants were noted as completed the study while only 22 attended the 12-month follow-up. This seems to me inconsistent, as only patients with obtained data at all time-points completed the study.
  • Line 304: At the first two follow-up visits, the RDs were more likely to fulfill the 24H-R. What was the reason for this? More RD or patient related?
  • When comparing the final results in energy and nutrient intake among participation level the authors did not observe any difference except for proteins. Why was this not shown per age-group (e.g. 0-2, 2-5, 5-21), tumor type or cancer evolution as all this data is available in the current study and it seems not expected that differences in dietary intake originate from participation levels. This should be recalculated and presented in the manuscript.
  • Also, for other tables comparisons per age-group or cancer diagnosis type would be highly relevant.
  • Some of the table footnoted are incorrect or partially missing.
  • Lastly, suggestions for improved feasibility for follow-up nutritional intervention studies should be provided as discussion.

Reviewer 3 Report

I read the article with great interest, because in my opinion the topic is very important and frequently underestimated.

The clinical approach and the involvement of parents is of great importance in this setting and I completely agree with the authors about the need of early intervention and regular follow-up for pediatric cancer patients, because of the risk of metabolic diseases in the survivor's population. I found useful to test the feasibility of the program in order to define a standard of nutritional intervention, simple to apply in daily clinical practice. Good job.

Round 2

Reviewer 1 Report

The authors have improved their study substantially

I have some remaining questions:

  • In the results section 3.1 the eligibility rate and recruitement rate are described. 109 patients (74%) were eligible. However, after this another 3 (because of not speaking french) and 14 (because they were contacted too late) patients were excluded. Then, the recrruitement rate was calculated with the remaining 92 patients. However, the 17 (3 + 14) pts should be somewhere: in my opinion in the eligibility numbers.
  • Line 522, discussion section: it states that vitamin C level is increased after the study. However, this is only depicted in the table, and is not significant (p 0.08). Should either be discussed why the authors think it a relevant result, or removed from the discussion (my preference)
